# Health-Related Quality of Life Assessments by Children and Adolescents with Sickle Cell Disease and Their Parents in Portugal

**DOI:** 10.3390/children9020283

**Published:** 2022-02-18

**Authors:** Clara Abadesso, Susana Pacheco, Maria Céu Machado, G. Allen Finley

**Affiliations:** 1Hospital Prof. Dr. Fernando Fonseca, EPE—Amadora, 2720-276 Amadora, Portugal; susana.r.pacheco@hff.min-saude.pt; 2Pediatrics, Faculdade de Medicina, Universidade de Lisboa, 1649-028 Lisbon, Portugal; machadomariaceu@gmail.com; 3Anesthesia & Psychology, Dalhousie University, Halifax, B3H 4R2 NS, Canada; allen.finley@dal.ca

**Keywords:** HRQL, Health Related Quality of Life, Sickle Cell Disease, PedsQL, Pediatric Quality of Life Inventory^TM^, children, adolescent

## Abstract

Health-Related Quality of Life (HRQL) can be used to measure the impact of Sickle Cell Disease (SCD) on the child and their family and is generally reduced. No research has yet measured HRQL in Portuguese pediatric SCD patients. Objectives: (1) Describe and compare HRQL of children with SCD reported by them and their parents; (2) Compare with a pediatric population with no SCD; (3) Find predictive factors of HRQL in SCD children. Methods: Descriptive, case-control study that included sixty-eight children and adolescents with SCD (aged 3 to 18 years) and their parents. Control group—children with no SCD, matched by age, gender and ethnic background. HRQL was assessed using the multidimensional self-report PedsQL^®^ 4.0 Generic Scales. Summary scores for overall HRQL and subscale scores for physical, emotional, social and school functioning were compared within groups (children-parents) and with the control group. Clinical and socio-demographic variables were analyzed to find predictive factors of HRQL in pediatric SCD patients. Results: Children with SCD and their parents had significantly lower overall and all subdomains of HRQL, compared with the control group. Children with SCD also rated lower when compared with their parents (only significant for social functioning), with low to moderate correlations. Children and parent reports declined with increasing age. Higher pain frequency was associated with worse total and psychosocial domains of HRQL. The number of hospitalizations was a predictor of worse school score, and female gender was a predictor of worse emotional score. Conclusions: SCD significantly affects children’s HRQL. Parents can provide a good proxy report, although both evaluations are beneficial. Disease status, like number of hospitalizations and frequency of pain, influences HRQL. Interventions in SCD should consider improvements in HRQL as an important outcome.

## 1. Introduction

Sickle Cell Disease (SCD) is one of the most common genetic disorders in African descent population and is characterized by chronic hemolytic anemia, recurrent painful acute episodes and multiple complications. It requires acute care visits, frequent hospitalizations and contributes to increased morbidity and early mortality [1].

Hemolysis causes variable degrees of anemia that contribute to fatigue and activity limitations, and sickle cell pain episodes cause acute and chronic pain [2].

Pain is the most common acute complication in SCD children and chronic complications become increasingly prevalent in adolescents and young adults [2]. The acute and chronic complications lead to predictable disease-related symptoms that significantly impact patient’s well-being [3].

The patient’s perspective on health and well-being has become an accepted method to determine the impact of the disease and its treatment [4], as a Patient-Reported Outcome (PRO), and it has been widely used in clinical practice and research [4,5].

Health-Related Quality of Life (HRQL), a type of PRO, is defined as an individual subjective perception of the level of functioning and physical, emotional and social well-being. It can provide an assessment of how an illness, its complications and treatment affect the patient on daily functional outcomes. It is increasingly recognized as a valuable measure [4,6].

To fully assess a child’s HRQL, both parent-reported and child-reported HRQL are needed [7]. This provides a better understanding of the burden of the disease experience and guides these patients’ healthcare [3].

The Pediatric Quality of Life Inventory (PedsQLTM) Generic Core Scales PedsQL 4.0 is one of the best known and widely used generic HRQL questionnaires for children and adolescents [3,8].

In the past 15 years, studies examining HRQL in SCD have increased. A systematic review of HRQL in children and adults with SCD using generic HRQL questionnaires shows a significant impairment in HRQL [8]. Most studies also support that pain and other SCD-related complications have a negative impact on HRQL [8].

Only a few studies of HRQL in SCD pediatric populations were conducted outside USA [8,9,10,11,12].

SCD prevalence studies are lacking in Portugal. This disease is rather frequent in areas corresponding to ancient and recent immigration zones (mainly the south and urban areas). In the last 30 years, the number of immigrants from Portuguese speaking African countries has increased, and there still are high rates of mobility in and out of the country. This contributed to increasing the numbers of SCD patients in Portugal in recent decades. Estimates suggest around 1500 SCD patients in the whole country [13].

The average life expectancy of SCD patients is increasing. New therapeutics, better follow up and prevention of complications have been responsible for this outcome. The number of children being born in Portugal with SCD has also been increasing. In this context, it is important to access HRQL in this population. It is also crucial to improve the awareness of HRQL assessment among professionals and population, applying to our reality standards already validated in other realities.

There is no research measuring HRQL in pediatric SCD in Portugal.

The primary objective of this study was to describe HRQL (comparing patient and parent report) in a group of SCD children and adolescents and to compare it with a group of children with no SCD. The secondary objective was to determine possible associations and factors related to the HRQL in these SCD children.

Our hypothesis was that children and adolescents with SCD rate worse HRQL, as well as their parents, when compared to children with no SCD. We hypothesize that pain could be one factor to interfere with the HRQL.

## 2. Materials and Methods

### 2.1. Study Design

This was a descriptive case-control study.

### 2.2. Study Population and Selection of Samples

The Study Group included children with SCD (SCD Group) aged between 3 and 18 years from a pediatric hematology consultation at a hospital in the Lisbon suburbs. The control group included children with no SCD or chronic pain condition, recruited in other pediatric hospital consultations, and matched for age, sex and ethnic background.

All participants were recruited at scheduled hospital visit and an informed written consent was obtained by the research team members before completing assessments.

### 2.3. Measures and Procedures

The HRQL was assessed using the Pediatric Quality of Life Inventory™ Version 4.0 Generic Core Scales (PedsQL™ 4.0)—Portuguese version [14].

It consists of 23 items that assess four domains: (1) Physical functioning (8 items), (2) Emotional Functioning (5 Items), (3) Social Functioning (5 Items), and (4) School Functioning (5 Items).

Child self-report (CR) includes age groups 5–7, 8–12, and 13–18 years. The parent proxy report (PR) includes ages 2–4 (toddler), 5–7 (young child), 8–12 (child), and 13–18 (adolescent) and assesses parent’s perceptions of their child’s HRQL. The items of each form are essentially identical, differing in developmentally appropriate language and first or third person. The instructions ask how unpleasant each item has been during the past month. A 5-point Likert response scale is used across child self-report for ages 8–18 and parent proxy report (0 = never a problem; 1 = almost never a problem; 2 = sometimes a problem; 3 = often a problem; 4 = almost always a problem). Children aged 5–7 years use a simplified, 3-point rating scale (0 = not at all a problem; 2 = sometimes a problem; 4 = a lot of a problem), with each response choice anchored to happy, neutral and sad faces scale. Items are reverse scored and transformed to a 0–100 scale, so that higher PedsQL scores indicate better HRQL [15,16,17]. Complete case analysis was used.

The Physical Health Summary Score (8 items) and the Physical Functioning Scale are identical. To create the Psychosocial Health Summary Scale (15 items), the mean was computed as the sum of the items divided by the number of items answered in the Emotional, Social and School Functioning.

The PedsQL questionnaire was completed independently and separately by parents and children.

Socio-demographic data was obtained from a study-specific questionnaire (family information form) completed with the child’s caregiver information. It included child academic data, family background, parental education and occupation, marital status and monthly income.

Clinical data were obtained from the clinical file and included: SCD complications, total hospitalizations, number of acute care visits and number of hospitalizations in the preceding year. Pain frequency (using a Likert scale) was measured by children and parents. A multidimensional evaluation of pain in the SCD children and adolescents was done simultaneously [18].

There was user agreement from Mapi Research Trust (https://mapi-trust.org) (accessed on 15 March 2013). The study was approved by the hospital ethical committee.

### 2.4. Statistical Analysis

Descriptive statistics and frequencies were calculated for each group. Mean values and standard deviations (SD) are presented for continuous variables and proportions for categorical variables. Descriptive statistics for the total score, summary scores and subscores of PedsQL were computed (mean ± SD).

Chi-square analyses (for categorical variables) or independent *t*-test (for continuous variables) were used to address demographic differences between groups.

A comparison of SCD and control group HRQL scores was made using *t*-student test.

A paired *t*-test was used to calculate the difference between the mean scores of the child and parent. Pearson correlation coefficients were used to determine the association between child and parent HRQL scores. Correlation coefficients <0.3 were considered poor agreement, between 0.3 and 0.5 moderate agreement and >0.5 strong agreement.

Pearson’s correlation analyses were used to test associations between socio-demographic and clinical variables and all PedsQL scores.

One-way ANOVA was conducted to determine the relationship between the categorical variables (age groups and pain frequency) in the child and parent PedsQL scores.

Multiple regressions were used to identify significant predictors of the total and subscores of PedsQL in subjects with SCD. Variables included: pain frequency and number of hospitalizations (those that were statistically significant in univariate analysis). Because age-related differences in scores are likely to occur, we attempted to account for this possibility by including age as a covariate in regression models when testing for significant predictors of PedsQL scores.

Data analysis was conducted using Statistical Package for the Social Sciences version 20.0 (SPSS 20.0). Significance levels were set at *p* < 0.05.

## 3. Results

### 3.1. Patient Characteristics

SCD group included sixty-eight children and adolescents. For children above 5 years a total of 57 parent–child pairs were studied; two reports were only answered by the child and one was only answered by parents (Figure 1).

The control group included 68 children and adolescents (60 dyads child–parent). These were recruited from other consultations: allergology (33%), surgery (7%), general pediatrics (30%).

For children aged 2–4 years there are only parent-report questionnaires (8 for each group).

Table 1 shows the child’s socio-demographic characteristics and family socio-economic status for each group. There was a significant difference between groups in school performance (failed years and need of school support). No difference was found in socio-economic status between groups.

All children (study and control group) had an African background. Most of the parent-report questionnaires were completed by the mother (n = 55) in the SCD group.

Disease-related characteristics of the SCD patients are listed in Table 2.

### 3.2. Comparison of HRQL between SCD and Non-SCD Patients

Children with SCD had worse HRQL in all summary scores than children without SCD, both in CR and PR (Table 3).

### 3.3. HRQL Scores Child–Parent Comparisons

Children with SCD rated their HRQL lower than their parents, but the difference was not significant, except for social functioning (*p* = 0.047). This difference was related to the age group 5–7 years with a significant difference between child and parent mean social score (77.5 ± 16.9 versus 90.3 ± 12.6; *p* = 0.015).

There was no significant difference between parents and children in the control group.

In the SCD group, Pearson’s correlations between parent and child reports showed low-moderate correlations (R between 0.30 and 0.43) (Table 4). There was no significant correlation in the emotional score.

In age groups 5–7 years and 8–12 years, only school functioning had a significant correlation (0.48 and 0.51, respectively). The teenage group showed higher and significant correlations in all domains, except for school functioning (range: 0.46–0.53).

### 3.4. Age and Gender Influence

There was a progressive decline of all CR and PR scores with increasing age in the SCD group (Figure 2), significant in parent-reported total score (*p* = 0.027) and physical score (*p* = 0.025) but not for child-reported total score (*p* = 0.344).

Differences between both groups increased with age, being more significant and for all scores (both child and parent) for the 13–18 year-old group.

In the SCD group female gender was associated with statistically significant lower CR emotional functioning (female: 59.8 ± 23.9; male 72.5 ± 18.1; *p* = 0.023). No other gender differences were found.

### 3.5. HRQL and Clinical Variables (SCD Group)

There was a decline in all child- and parent-reported scores with the number of hospitalizations. Child-reported total scores were 69.9 ± 12.3; 66.6 ± 17.9; 46.0 ± 13.7 (*p* = 0.042), for 0 (n = 29), 1–2 (n = 27) or ≥3 (n = 3) hospitalizations (in the previous year), respectively. Parent-reported total scores were 73.7 ± 15.2; 67.4 ± 15.7; 51.9 ± 19.9 (*p* = 0.024), for 0, 1–2, or ≥3 hospitalizations, respectively. There was also statistical significance in the decline of child-reported social (*p* = 0.013) and psychosocial (*p* = 0.032) scores and parent-reported physical (*p* = 0.01) and school (*p* = 0.021) scores, with increasing number of hospitalizations.

The 3 children with a previous stroke showed lower partial and total CR and PR scores. Children with severe sequelae (n = 5) had significant lower CR total (68.3 ± 15 vs. 54.3 ± 19.8; *p* = 0.05), school (66.3 ± 17 vs. 44 ± 26.7; *p* = 0.01) and psychosocial (71.8 ± 14.9 vs. 56.3 ± 15.4; *p* = 0.031) scores.

The higher the child-reported frequency of pain episodes, the lower CR physical, social and total score and PR physical score (*p* < 0.05). In addition, the higher the parent-reported pain episodes the lower the CR physical, emotional, social, psychosocial and total score (*p* < 0.05). These results are shown in Table 5.

Hospitalizations (total number and previous three years) showed an inverse correlation with CR total score (−0.322; *p* = 0.013), psychosocial score (−0.341; *p* = 0.008) and social and school score.

The number of hospitalizations during the previous year had an inverse correlation with CR social score (−0.267; *p* = 0.041).

An inverse correlation was found between pain frequency and CR physical score (−0.40; *p* = 0.003), social score (−0.339; *p* = 0.013) and total score (−0.342; *p* = 0.012).

### 3.6. Predictive Factors of HRQL (SCD Group)

In the multiple regression analyses, significant predictors of total and summary scores of HRQL were identified (Table 6).

Higher pain frequency was the dominant variable associated with lower total, physical, psycho-social, emotional and social child-reported PedsQL scores. Higher number of hospitalizations (during the previous 3 years) and school performance were associated with lower school score. The female gender was a predictor of lower emotional scores.

## 4. Discussion

The present study shows that Portuguese children with SCD and their parents rated total and all subdomains of HRQL lower than children without this disease and their parents, as found in previous studies in SCD pediatric populations [2,7,17,19,20,21,22,23]. As an exploratory study, it also identifies factors associated with worse HRQL (pain frequency, number of hospitalizations, gender).

Compared to other studies which used PedsQL: in our study, the CR total score was similar, but the CR physical score was significantly lower than other studies [2,22,24]. For example, in the Dampier C et al. study (N = 1392) [2], CR total score was 72.28 ± 16.02 and CR physical score was 74.34 ± 18.47, both being statistically different from our results (*p* < 0.05 and *p* < 0.005, respectively). These results reinforce that impaired HRQL, both in adults and children, is especially noted in physical functioning domains [8]. Our lower CR physical score could be related to multiple variables like disease status.

In our sample, parents reported significantly higher in total and most of the other scores, than some previous studies [22,24], related mostly to a higher PR social score.

The significantly lower HRQL scores are comparable to or worse than other chronic diseases, such as diabetes, gastrointestinal, rheumatologic, cardiac conditions, asthma, obesity, end-stage renal disease, cancer and epilepsy [6,8,25,26]. The severity of impairment of multiple domains of HRQL is especially evident in chronic pain conditions [6].

Both parent-reported and child-reported HRQL are needed to fully assess a child’s HRQL [7], giving a more complete understanding of the impact of SCD on the child and their family. Parent reports may be used to gain a more complete picture of HRQL but should not substitute self-reports. Proxy reports are useful when children are unable to complete measures themselves due to young age, physical illness, emotional distress or cognitive impairment [27]. Even if differences exist, they may provide a more complete picture of child HRQL [27,28]. It is recommended that patient self-report be the primary measure of HRQL and parent proxy report a secondary measure [1].

Parent-proxy reports are often lower than child self-reports. Parents of children who are chronically ill generally report poorer HRQL than do children [28]. In previous studies, children with SCD consistently reported their HRQL to be better than that described by their parents, with a statistical difference in most domains [7,17,20,22].

The present study shows that in the SCD group, PR scores were higher than CR scores, although without statistical significance, except for social functioning in the age group 5–7 years. So, in the present sample, we could say our parents seemed to be a good proxy report of their child’s HRQL, although they are not so aware of real social functioning in smaller children.

As reported from the Comprehensive Sickle Cell Centers Clinical Trial Consortium, there is a decrease in correlation between parents and children at lower ages, and the less easily observed and more internal items (e.g., those related to social and emotional functioning) showed lower parent–child correlations [2]. The same has been observed in other HRQL studies in children with a chronic illness [28]. In our sample we also observed that child–parent correlation was moderate in the physical domain and low in the psychosocial domains, with no correlation in the emotional domain. In addition, these correlations were higher in older children.

Agreement might be expected to increase with the child’s age, especially since greater verbal skills may facilitate children’s abilities to describe their experiences and emotions to parents [28].

This study’s mean total and physical scores (for both CR and PR report) were slightly more than one standard deviation (SD) below the healthy population. Varni et al. [15,16] proposed that children with a chronic condition with scores that are one SD or more below the child population mean are “at risk” for impaired HRQL. Using this criterion 50.8% of children with SCD would be at “risk” for low overall HRQL, in the present sample.

The SCD group showed a progressive decline of all CR and PR scores with increasing age. In the age group 2–4 years old, the only significant difference between SCD and control group was in school score (parent report). This means the impact in school performance in SCD is seen very early. The difference between groups scores increased with age. It seems that, as the disease progresses the impact on physical and psychosocial health, from both the child and the parent perspective, becomes more significant. Adolescents were significantly more impaired in all domains, especially for physical and social functioning, than young children. This progressive decrease of CR and PR scores with age has been previously reported [2,19].

It is expected that HRQL decreases with increasing age, as there are well recognized age-related rises in frequency and prevalence of acute and chronic complications. By age-specific developmental processes the adolescents may be more aware of the impact of the illness on their physical and emotional well-being. In addition, adolescents may be exposed to a greater number of activities than children, highlighting their disability [2,6].

Besides this decline of HRQL with age, this variable was not a significant predictor in our regression analysis.

Gender differences have also been found in previous HRQL studies, where girls had lower scores than boys concerning total, physical and emotional functioning [2,19,20]. Our study just showed a significant difference in the emotional functioning, where girls had lower scores than boys, and female gender was a predictive factor for worse emotional functioning.

Our study showed that child-report and parent-report school functioning positively correlated with school performance. Furthermore, child-report school score had a weak inverse relationship with the number of ED visits in the previous year. Difficulties in school may be due to several factors: missing school days because of the burden of the disease (acute complications, hospitalizations, ED visits, routine visits, etc.) and the neurological impact of the disease. Studies of the cognitive aspects of SCD have indicated small IQ deficits, which probably suggest subtle neurological effects due to subclinical cerebral infarcts [29]. These children/adolescents have lower school performance due to multifactorial issues.

Most studies support pain and other SCD-related complications as having a negative impact on HRQL [8].

Although multiple linear regression models do not show a high R2 determination coefficient, some variables have a clear association in our study. The frequency of pain episodes was the variable most associated with worse HRQL scores. The female gender was a predictor of worse emotional functioning. School performance was a predictor of school score. The number of hospitalizations (during the previous 3 years) was a predictor of the social score.

The association of more SCD pain events and the lower quality of life has also been reported in adults [21] and children [2,8,30,31]. In childhood, recurrent pain is marked by both physical and psychosocial difficulties [31].

Other pediatric SCD research also found an inverse correlation between the number of hospitalizations and ED visits and overall HRQL and most of the domains [2,20,22]. In fact, pain frequency and number of hospitalizations can reflect acute complications or disease activity that require treatment in the ED or the hospital and may have a negative impact on HRQL. In addition, pain in children with SCD is associated with increased distress [3].

Previous studies associate worse HRQL with disease severity (whose criteria were: overt stroke, acute chest syndrome, three or more hospitalizations for painful events in the prior 3 years) and disease related complications (occurrences of pain, priapism and avascular necrosis of hip/shoulders) [2,3,7,17,32,33]. We analyzed each criterion of disease severity separately (pain frequency, the number of hospitalizations, previous stroke or important sequelae) and those with more severe disease had worse HRQL. However, no global severity instruments have been validated [22].

Most of the previous studies in the SCD pediatric population used a generic HRQL instrument (most of them used the PedsQL). Recently, a pediatric disease-specific module, PedsQL^TM^ SCD-specific Module (1), has been developed. This module increases the specificity of HRQL measurement in these children.

In the PedsQL Sickle Cell Disease (SCD) Module, the pain and hurt and pain impact scales demonstrated the strongest measurement properties for child self-report [33]. Using PedsQL SCD Pain and Hurt and Pain Impact scales, the scores were compared with disease severity and the PedsQL Generic Core Scale. Children with more pain had lower HRQL scores related to pain.

In summary, in comparison to previous studies, our results suggest some similarities and some differences in the HRQL CR and PR assessments. Our children rated similar or lower than in other studies (specially in physical functioning); our parents rated higher than in other studies. There was no statistical difference between PR and CR scores (as in previous studies), except for social functioning in the age group 5–7 years, but parents rated higher than children.

In this study our data is not enough to explain these differences in PR and CR scores compared to other studies, but we could infer that it can be related with multiple variables like disease status, health care support, school support, socio-cultural and economic aspects, that we cannot compare with other samples.

We also observed decline of CR and PR scores with increasing age and gender difference, but only in emotional functioning, and the impact of the pain and hospitalizations in HRQL in these children, like in previous studies.

### 4.1. Implications of the Study

Incorporating HRQL measurements into clinical practice allows the health care team to identify areas of impaired function that can help determine which interventions (medical, psychological and behavioral) could be developed to address specific needs of each child with SCD.

A comprehensive evaluation and a multidisciplinary intervention are needed to best manage the health problems (physical, mental and social) and improve overall care for patients with SCD. Interventions in SCD should consider improvements in HRQL as important outcomes [21].

Through health education and health promotion strategies aimed at helping patients take control of their condition and identify social support, these care providers play a key role in helping to better manage the SCD [20].

In addition, as pain is one of the factors most commonly being associated with the HRQL, increased attention to pain management and further inquiry for child’s pain experience should be addressed to improve HRQL.

### 4.2. Strengths of the Study

The majority of patients in the present study have HbSS, the most severe form of SCD. Patients with hydroxyurea or chronic transfusion treatment were not excluded, as in some previous studies [20]. Both child and parent report HRQL were assessed. Proxy evaluation adds complementary information to HRQL child report. The enrollment of children and adolescents with SCD in our study was done in the outpatient setting when they are usually in a steady state. Race, gender and age matched with the control group, intended to avoid the association of the HRQL and socio-economic status.

### 4.3. Limitations of the Study

The study’s main limitations include: 

(1) Cross-sectional design, small sample and patients from only one center.

(2) A generic rather than a disease-specific instrument for HRQL evaluation was used. The SCD-specific instrument has been validated for SCD, but the Portuguese language validated version was not available at the time of this study.

(3) The lack of a specific disease severity classification limits this correlation.

## 5. Conclusions

This study extends the current literature showing that children and adolescents with SCD have significantly lower HRQL when compared with children with no chronic pain condition, having lower scores in all subdomains. These findings are similar to other SCD pediatric populations. Parents were a good proxy report in this population, except for social functioning where children reported significant lower scores. Pain frequency was the most significant predictor for HRQL scores. The number of hospitalizations was a predictor of lower school scores and gender was associated with the emotional score.

The use of a disease-specific instrument rather than a generic one may improve the HRQL assessment in SCD children. There is a real need to offer a multidisciplinary approach and to provide psychosocial and school interventions to improve these children’s HRQL.

To our knowledge, the present study is the first to measure HRQL in pediatric SCD in Portugal.

## Figures and Tables

**Figure 1 children-09-00283-f001:**
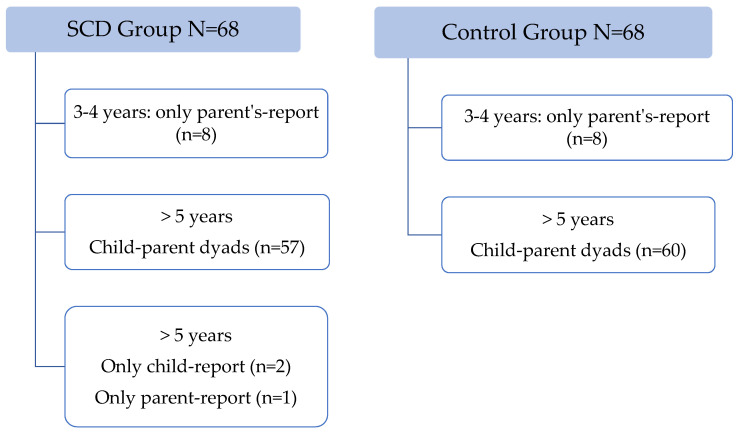
Patients included.

**Figure 2 children-09-00283-f002:**
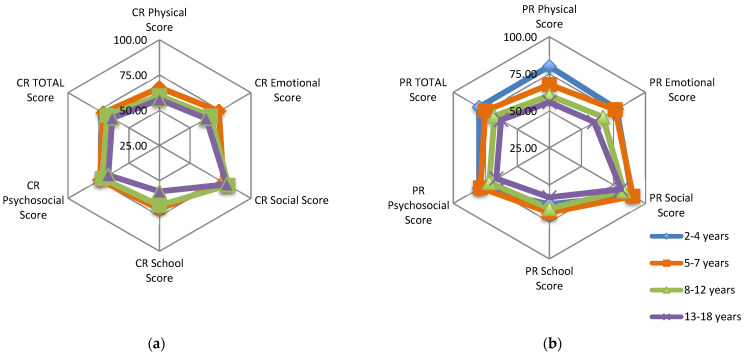
SCD Group: (**a**) Child-reported PedsQL mean scores for age groups: 5–7, 8–12, 13–18 years; (**b**) Parent-reported PedsQL mean scores for age groups: 2–4, 5–7, 8–12, 13–18 years.

**Table 1 children-09-00283-t001:** Demographic and socio-economic characteristics (SCD and Control group).

	SCD Group (n = 68)	Control Group (n = 68)	*p*
Child age (Mean ± SD)	10.13 ± 4.5	9.76 ± 4.11	ns
Age groups, N (%)			
3–4 years	8 (11.8)	8 (11.8)	ns
5–7 years	17 (25)	14 (20.6)	
8–12 years	20 (29.4)	27 (39.7)	
13–18 years	23 (33.8)	19 (27.9)	
Child gender			
male/female (%)	56/44	52/48	ns
School Failed years (M ± SD)	0.92 ± 1.15	0.39 ± 0.87	0.011
Need of school support, N (%)	14 (20.6)	2 (2.9)	0.001
Parents’ education < 9th grade			
Mother (%)	65	80	ns
Father (%)	55	77	
Unemployment			
Mother (%)	32	35	ns
Father (%)	24	31	
Monoparental families (%)	30	34	ns
Median total monthly income (euros)	694	850	ns

**Table 2 children-09-00283-t002:** Disease characteristics of the SCD Group (N = 68).

Variable	SCD Group (N = 68)
Hemoglobinopathy, N (%)	
SS	65 (95.6)
SD	1 (1.5)
Sβ + thalassemia	2 (2.9)
Disease-related complications *, N (%)	
Stroke	3 (4.4)
Acute Chest Syndrome	8 (11.7)
Bone necrosis (avascular)	4 (5.9)
Splenic sequestration	13 (19)
Other (Meningitis)	1 (1.5)
Number Hospitalizations (mean ± SD)	
Total	12 ± 10.37
Prior 3 years	4 ± 4.1
Due to VOC	6.7 ± 6.9
Hospitalizations in the previous year (%)	0 (47%), 1–2 (40%), ≥3 (13%)
Number ED visits (mean ± SD)	
Prior 3 years	7.1 ± 5.5
Previous year	2.3 ± 2.4

Abbreviations: SD: standard deviation; VOC: Vasoocclusive crisis; ED: Emergency Department. * Disease sequelae: Hemiparesis (2), Hip arthrosis (2), visual deficit (1); leg ulcer (1).

**Table 3 children-09-00283-t003:** HRQL in SCD group and Control group.

	SCD Group Mean (SD)	Control Group Mean (SD)	*p* Value
Parent-proxy report			
Total Score	69.5 ± 16.4	83.2 ± 10.45	*p* < 0.001
Psychosocial Health	73 ± 16.2	81.5 ± 10.14	*p* < 0.001
Physical Health	63.2 ± 20.9	86.4 ± 15.8	*p* < 0.001
Emotional Functioning	68.4 ± 21.9	76.8 ± 16.2	*p* = 0.012
Social Functioning	85 ± 14.2	94 ± 9.9	*p* < 0.001
School Functioning	64 ± 20.8	73.4 ± 16.8	*p* = 0.005
Child self-report			
Total Score	67.14 ± 15.8	81.7 ± 9.5	*p* < 0.001
Psychosocial Health	70.5 ± 15.4	80.5 ± 10.7	*p* < 0.001
Physical Health	60.8 ± 20	84 ± 12.5	*p* < 0.001
Emotional Functioning	67.3 ± 21.4	76 ± 17.7	*p* = 0.018
Social Functioning	79.8 ± 17.2	90.2 ± 12.7	*p* < 0.001
School Functioning	64.4 ± 18.7	75.4 ± 16.5	*p* = 0.001

**Table 4 children-09-00283-t004:** Pearson correlation coefficient between child and parents.

	SCD Group (N = 57)		Control Group (N = 60)	
PedsQL Scores	R	*p*	R	*p*
TOTAL	0.42	0.001	0.527	<0.001
Psychosocial	0.345	0.009	0.529	<0.001
Physical	0.435	0.001	0.517	<0.001
Emotional	0.234	0.08	0.588	<0.001
Social	0.306	0.021	0.441	<0.001
School	0.354	0.007	0.545	<0.001

**Table 5 children-09-00283-t005:** HRQL and pain frequency—child-reported (SCD Group).

	<1 × Month (n = 30)	1–3 × Month (n = 15)	2–6 × Week (n = 7)	Everyday (n = 1)	*p* (One-Way ANOVA)
Child-Report					
Total score	71.6 ±12	61.3 ± 18.2	56.5 ± 21.9	54.3	0.048
Physical score	67 ± 17.3	51.5 ± 23.2	48.6 ± 19.2	56.25	0.037
Emotional score	71.6 ± 20.3	64.6 ± 23.1	52.1 ± 26.2	60	0.198
Social score	83.9 ± 13.7	76.8 ± 15.9	63.5 ± 25.4	70	0.03
School score	66.8 ± 16	58.3 ± 21	66.4 ± 25.7	30	0.167
Psychosocial score	74.1 ± 11.8	66.5 ± 16.9	60.7 ± 23.8	53.3	0.097
Parent-Report					
Total score	71.4 ± 13.5	63.1 ± 17.6	58.7 ± 19	87.5	0.088
Physical score	65.6 ± 17.9	56 ± 21	43.6 ±23.1	92.8	0.02
Emotional score	71.1 ± 16.1	58.9 ± 25.5	55.5 ± 32.1	100	0.071
Social score	84.9 ± 12.5	85 ± 14.6	77.1 ± 20.5	100	0.396
School score	67.3 ± 17.6	57.1 ± 22.3	67.5 ± 23.6	33.3	0.172
Psychosocial score	74.4 ± 13	67 ± 17.4	66.7 ± 22.2	84.4	0.318

**Table 6 children-09-00283-t006:** Summary of multiple regression analysis.

Score	Pain Frequency		Hospitalizations (N)		Female Gender		School Performance		Adj R^2^	*p*-Value
	B (SE)	*p*-Value	B (SE)	*p*-Value	B (SE)	*p*-Value	B (SE)	*p*-Value		
Total	−4.542 (1.935)	0.023	−0.332 (0.193)	0.090	-	-	-	-	0.166	0.008
Physical	−5.475 (2.536)	0.036	−0.269 (0.252)	0.291	-	-	-	-	0.099	0.044
Psychosocial	−4.043 (1.885)	0.037	−0.367 (0.187)	0.056	-	-	-	-	0.166	0.008
Emotional	−5.377 (2.594)	0.043	-	-	12.744 (5.741)	0.031	-	-	0.159	0.009
Social	−5.021 (2.045)	0.018	−0.373 (0.203)	0.073	-	-	-	-	0.169	0.007
School	−1.369 (2.454)	0.58	−1.373 (0.503) *	0.009	-	-	5.341 (2.398)	0.032	0.282	0.002

Variable age included in all regressions; * Number of hospitalizations during the previous 3 years.

## Data Availability

The data presented in this study are available on request from the corresponding author.

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
