# Peer review of "Health-Related Quality of Life Assessments by Children and Adolescents with Sickle Cell Disease and Their Parents in Portugal"

_children, 2022, doi:10.3390/children9020283_

Round 1

Reviewer 1 Report

This paper by Dr. Abadesso and colleagues provides the results from a quality-of-life survey of children with SCD and their parents in Portugal. It is generally well performed. There Is not a lot that is new compared to the many previous papers on this topic, but it will represent a useful addition to the literature. I have only a few minor comments for revisions.

  1. The title does not capture the true focus or full impacts of the paper. I suggest changing it to Health-Related Quality of Life assessments by children and adolescents in Portugal with Sickle Cell Disease and by their parents.
  2. The blue color in Fig. 1 should be changed (likely to a lighter more transparent color) to make the writing more visible.
  3. Graphic 1 needs more explanation.
  4. The authors should expand the Discussion to make it more clear how the current results differ from and/or complement those of prior studies.

Author Response

Thank you for your comments/suggestions. Reply is written in red:

  1. The title does not capture the true focus or full impacts of the paper. I suggest changing it to Health-Related Quality of Life assessments by children and adolescents in Portugal with Sickle Cell Disease and by their parents. – we introduced a modification of the title as requested
  2. The blue color in Fig. 1 should be changed (likely to a lighter more transparent color) to make the writing more visible. – Done
  3. Graphic 1 needs more explanation. – this graphic shows the progressive decline of the scores with increasing age (it is explained in the text)
  4. The authors should expand the Discussion to make it more clear how the current results differ from and/or complement those of prior studies. – We introduced some edits to the discussion to improve /make more perceptible the comparison with other studies. Our data cannot explain some differences, but we will focus on these issues in subsequent research.

Reviewer 2 Report

It would be nice in discussion to comment the results not only repeat them again. In lines 308 - 313 authors presented slightly different results from others. What the is the commentary?

Author Response

Thank you for your comments/suggestions! Answer to the comments in red:

  1. It would be nice in discussion to comment the results not only repeat them again. In lines 308 - 313 authors presented slightly different results from others. What the is the commentary? - We introduced some edits to the discussion to improve /make more perceptible the comparison with other studies. Our data cannot explain some differences, but we will focus on these issues in subsequent research.